# Chromium (VI) Adsorption on Modified Activated Carbons

**Alicja Puszkarewicz \***  **and Jadwiga Kaleta**

Department of Water Purification and Protection, Faculty of Civil and Environmental Engineering and Architecture, Rzeszow University of Technology, 35-959 Rzeszów, Poland
\* Correspondence: apuszkar@prz.edu.pl

**Abstract:** The paper presents the research of adsorptive properties of activated carbons to chromium (VI) removal from the water solution. Different types of carbon were investigated: WD-ekstra (WDA), WD-ekstra modified by salt acid WD(HCl) and nitrogen acid WD(HNO$_3$). The adsorption kinetics, influence reaction, temperature and kind of carbon were determined using static methods. The adsorption of chromium (VI) onto all activated carbons was most efficient at pH 2. The time of adsorption balance for carbon WD(HCl) was 2.5 h and for carbon WDA 4.5 h. The influence of carbon modification and temperature on the effectiveness of adsorption of hexavalent chromium was determined on the basis of the Freundlich adsorption isotherm. The research showed that the adsorption capacity of Cr(VI) increased with the increasing temperature of the solution. The most efficient adsorbent in static conditions was found on WD-ekstra modified by HCl. The adsorbent applied in dynamic conditions as the filtration bed (v = 2 m/h, time of retention $T_R$ = 420 s, initial concentration $C_0$ = 1 mg/dm$^3$), it removed chromium (VI) from water very effectively. Adsorption capacities after exhaustion of the bed were obtained at $P_{eWD \ (HCl)}$ = 4.35 mg/g. On the basis of executed research, chromium ions were successfully eliminated onto modified carbon WD(HCl) that makes its employment capabilities in the systems purification of water.

**Keywords:** adsorption; modified activated carbon; removal of chromium(VI); Freundlich isotherms; filtration

## 1. Introduction

Chromium (VI) is considered to be a bioavailable element due to its good solubility in water, strong oxidation properties and permeability through cell membranes. Unfortunately, chromium (VI) compounds cause toxic, mutagenic and carcinogenic effects on individual organisms [1]. It is more and more often observed that their permissible concentration in surface waters is exceeded several times, which is why it is extremely important to search for effective and economical methods of removing them.

Sorption is one of the more effective and more frequently introduced methods [2,3]. An additional advantage of this method is the possibility to burn a sorbent (with the addition of alkali) and further processing of the adsorbed chromium [4]. This process in a defined system (flow through or non-flow through) lasts until the equilibrium concentration is reached. The time of the equilibrium reaching depends, among others on the adsorbate particle size and its susceptibility to the sorption process. In the state of dynamic equilibrium, there is a specific distribution of the adsorbate between the solution and the adsorbent. This distribution is described by means of adsorption isotherms showing the relationship between the amount of substance adsorbed by the adsorbent mass unit and the equilibrium concentration of the adsorbate at a constant temperature [5]. Among the adsorption isotherms, the equations of Freundlich and Langmuir are particularly useful for the mathematical

description of adsorption from dilute aqueous solutions. The Freundlich equation is commonly used, especially in relation to aqueous solutions [6], due to its simple construction, ease of mathematical solution and sufficient accuracy in the description of experimental results.

In the sorption process, the most important element is the sorbent used [7–10]. The most popular and effective sorbent for removal of heavy metals (including Cr(VI)) is activated carbon [11,12]. The basic reason for the use of fossil carbons as raw materials for obtaining adsorbents is the relatively easy formation of a developed porous structure in them. In order to obtain activated carbons with a broad distribution of pore volume as a function of their radius, it is recommended to use carbons with a low degree of metamorphism and a high content of volatile parts [13]. Carbon for adsorption from solutions is obtained mainly from peat, wood, brown, hard coal and many other materials, e.g., from Peganum harmala seed (PPAC), sunflower seed, and waste biomass [14–16]. The basic technological operations used to obtain activated carbons are carbonization and activation. Some types of activated carbons, additionally after the activation process, are subjected to various refinement, modifying operations or their surfaces are covered with catalysts [17–21]. The efficiency of chromium removal in the adsorption process is very different and depends on the type of carbon used, the method of modification and many other factors. Selomulya C. et al. [22] observed that the adsorption of Cr(VI) on various activated carbons is best at pH 2–4, and the time to reach the adsorption equilibrium does not exceed 6 h and the adsorption capacities are in the range of 60–75 mg/g. Karnjanakom et al. [23] report that activated carbon from Leucaen leucocephala (Lam.) De Wit and commercial activated carbon-UCS modified with cationic surfactants (HDTMA and DDAB) used to remove chromium (VI) ion significantly increased the adsorption capacity compared to the adsorbent without modification. Activated carbon (UCS-DDAB) showed a maximum adsorption capacity of 3.46 mg/g. In turn, Jain M. et al. [24] showed in their research that activated carbon prepared from sunflower head waste, modified with hydrochloric acid to increase porosity and additionally impregnated with $Fe_3O_4$, achieved an adsorption capacity of 4.4 mg/g.

This paper presents studies on the usefulness of WD-extra activated carbon and modified by $HNO_3$ and HCl for removal of chromium (VI) from water. Under static conditions, adsorption isotherms, pH effect and adsorption kinetics were determined. Adsorption capacities and filtration isoplanes were determined in dynamic conditions.

## 2. Materials and Methods

The tests were carried out on a model aqueous solution. It was prepared on the basis of tap water, pH: 7.0–7.2 pH, to which chromium (VI) were added in the form of potassium dichromate $K_2Cr_2O_7$. Initial concentration of chromium in water was:

- 　under static conditions from $C_0 = 1 \div 500$ mg/dm$^3$.
- 　under dynamic conditions $C_0 = 1$ mg/dm$^3$.

### 2.1. Materials

Three types of activated carbon were used for laboratory tests:

(a)　Activated carbon WD—extra—which in the further part of the paper will be called WDA
(b)　Activated carbon WD—extra modified with hydrochloric acid (HCl), called WD(HCl)
(c)　Activated carbon WD—extra modified with nitric acid $HNO_3$- marked as—WD($HNO_3$).

Carbon WD-exstra were purchased from "Gryfskand—Hajnówka"company. All chemical reagents were of analytical grade and purchased from "Chempur" company.

Table 1 shows the characteristics of the activated carbon used.

**Table 1.** Characteristics of activated carbon WD—extra.

| Indicator | Volume |
|---|---|
| Density bulk density, [g/L] | 390 ÷ 415 |
| Granulation, [mm] | 1 ÷ 1.5 |
| Specific surface, [m$^2$/g] | 950 ÷ 1050 |
| Aggregate volume of pores, [cm$^3$/g] | 0.85 ÷ 0.95 |
| Adsorption of iodine, [mg/g] | 900 ÷ 1000 |
| Dechloration capacity, [cm] | 4 ÷ 5 |
| Mechanical durability [%] | 90 |

### 2.1.1. Modification of Activated Carbon with Hydrochloric Acid HCl

Carbon WD—extra was dredged, dried, and then treated with hydrochloric acid (1:1) for 24 h. The next step was rinsing it with distilled water until the content of chlorides in the water was less than 10 mg/dm$^3$. After rinsing, the carbon was dried at 105 °C for 12 h. This operation was repeated three times.

### 2.1.2. Modification of Activated Carbon with Nitric Acid HNO$_3$

Modification of activated carbon WD—extra with nitric acid consisted in dredging, drying, treating with nitric acid HNO$_3$ (1:1) and bringing to a boil. The heating was led under the reflux condenser. Then, the filtrate was poured out, and the sample of activated carbon was again treated with acid. This operation was repeated three times. The obtained sample was washed with distilled water until pH 6.5 was reached. Then, it was dried at 105 °C.

### 2.2. Tests under Static Conditions

In order to determine the optimal conditions for the sorption process and the effect of carbon modification on the sorption of Cr(VI), the research work proceeded according to the following stages:

(a) determination of the solution pH and carbon modification effect on chromium sorption (pH 2–10), for $C_0$ = 10 mg/dm$^3$, carbon dose 1 g/dm$^3$, adsorption time 12 h.
(b) determination of the chromate ions sorption kinetics (for adsorption times from 0.25–12 h).

In this stage the research work was carried out in vessels with the adsorptive content of 0.5 dm$^3$, for $C_0$ = 10 mg/dm$^3$, carbon dose 1 g/dm$^3$.

(c) preparation of Freundlich adsorption isotherms for two selected carbons at three different temperatures 288.15 K (15 °C), 313.15 K (40 °C) and 338.15 K (65 °C) (for $C_0$ = 1–500 mg/dm$^3$ and carbon dose 1 g/dm$^3$).

### 2.3. Tests under Flow-Through Conditions

The tests were carried out on laboratory filters with 0.018 m diameter, filled with tested carbons, with gravitational flow from top to bottom and filtration speed of 2 m/h. The weight of activated carbon filling the filter was 40 g.

Sorption capacity at the point of penetration of the deposit (increase in chromium concentration in the spill) was determined from the dependence:

$$P = \frac{V \times (C0 - Ck)}{M} \tag{1}$$

*P*—sorption capacity of the deposit (mg/g)
*V*—volume of treated water (dm$^3$)

$C0$, $Ck$—initial and final concentration of chromium compounds (mg/dm$^3$)

$M$—mass of the deposit (g)

The control assay for Cr (VI) in treated water, was performed after filtering each 5 L of water.

The amount of chromium (VI) present in the water was analyzed by measuring the absorbance of the purple complex o Cr(VI) with 1.5- diphenylcarbazide at 540 nm spectrophotometrically.

## 3. Results and Discussion

### 3.1. Effect of pH

In order for the adsorption to proceed with a high intensity, the surface charge of the adsorbent and the electrical potential of the adsorbate should be opposite. For many solids, the potential-forming ions are H$^+$ and OH$^-$ ions, which entails that the surface potential can be influenced by changing the pH of the solution in which a given solid is found. In the case of a very strong adsorption of potential-forming ions on the surface of the solid (especially for oxides) it may happen that the charge of the adsorption layer will change its mark. This means that the surface of the solid body with which the adsorption layer is closely related will also change the sign. Therefore, depending on the pH of the solution in which the adsorbent is located, it will take up a positive or negative surface electric charge [5,25].

Within the neutral range, the inherently amphoteric character of the activated carbon surface has a positive surface charge. At a higher pH of the solution, the negative chromate ions significantly reduce their carbon-related affinity, as a negative surface charge may form on the surface of the particle and inside the carbon pores [26].

The solution pH effect on chromium (VI) adsorption is shown in Figure 1. For WD (HNO$_3$) the adsorption decreased with increasing the solution pH. For carbons WDA and WD (HCl), the reaction effect on chromium adsorption capacity was greater and definitely more beneficial at lower pH. Another work has also reported similar behavior using other adsorbents [22,24,27].

Further studies were carried out at pH = 2.

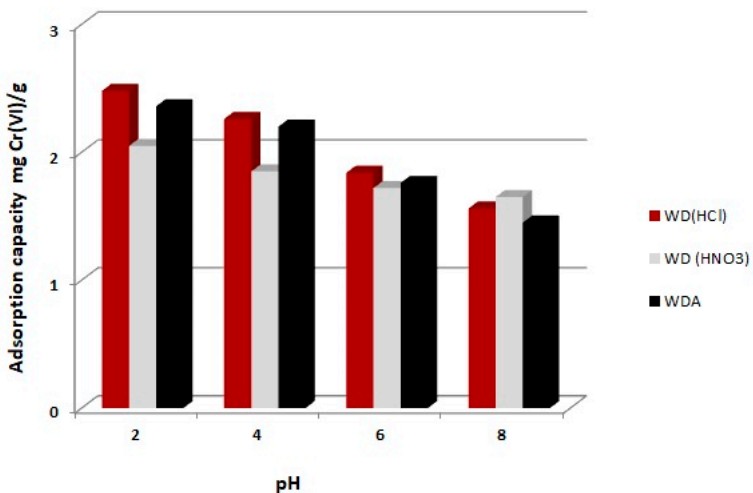

**Figure 1.** Effect of pH on the adsorption of chromium (VI).

### 3.2. Effect of Modification on Adsorption

The purpose of chemical modification of WDA carbon was to improve its sorption properties, because the modification of the carbon surface changes their structural and surface properties in a controlled manner [28]. Activated carbons due to a highly disordered crystal structure and large specific surface area can be easily chemically modified, among others as a result of the action of inorganic acids.

The tested carbon was treated with hydrochloric acid (HCl) and nitric acid ($HNO_3$). The specific surfaces of the tested carbons were determined by the BET method for $N_2$ adsorption at 76 K in liquid nitrogen (analyzer SORPTY 1750). The specific surface area of WDA was 1020 $m^2$/g, while for WD(HCl) it decreased (3%) to 990 $m^2$/g. The oxidizing nitric acid modification caused a significant reduction of the area to 740 $m^2$/g (30%).

The use of hydrochloric acid increased the adsorption capacity by approximately 5%, while the use of hot nitric acid (V) reduced this capacity by approximately 13% (Figure 2).

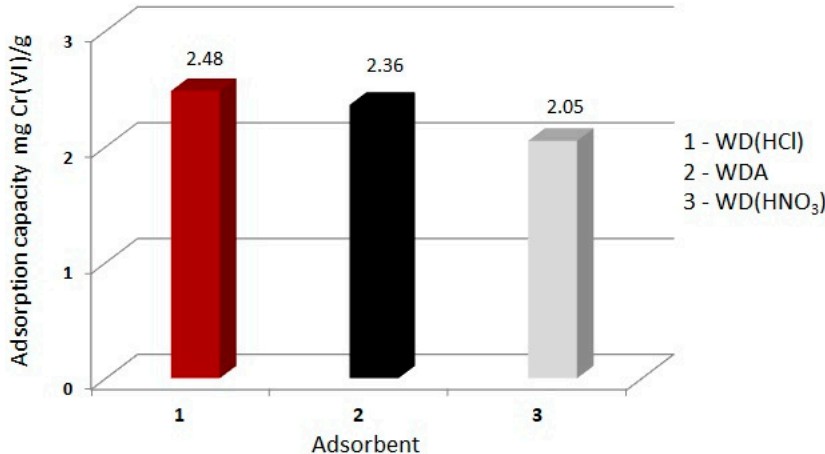

**Figure 2.** Influence of the carbon type on Cr(VI) adsorption capacity.

The positive effect of the hydrochloric acid (as a modifier) on the adsorptive properties of carbon was caused by its effect on inorganic admixtures contained in WD—extra carbon. The use of hydrochloric acid caused the ash removal of commercial carbon, decreased specific surface and thus increased the volume of mesopores in the carbon [28].

The effect of nitric acid did not result in increased adsorption capacity, most probably due to its strong oxidizing properties. Partial oxidation of carbon by such a strong oxidant may have contributed to the decrease in the specific surface area, through the destruction of the pore structure. In addition, treatment with such an oxidizer changes the chemical nature of the carbon surface [24]. Modification of the surface of activated carbons by oxidation increases the surface concentration of chemisorbed oxygen, thus increasing the polarity and hydrophilicity of the surface. This suggests the presence of oxidized carbon of various functional groups, e.g., phenolic, carboxylic, aldehyde on the surface. This may change the surface charge of carbon grains from positive to negative and weaken the affinity of chromate ions to the adsorbent [22,28].

Due to the adverse effect of carbon modification with nitric acid, WDA and modified WD (HCl) were used for further tests.

### 3.3. Adsorption Kinetics

The adsorbent saturation of each adsorbent particle depends on the rate of diffusion of particles absorbed into the interior of the granule. Adsorption kinetics makes it possible to determine the dependence of the adsorption rate on the adsorbent properties and the conditions of the adsorption process itself. Adsorption kinetics therefore depends on the type of adsorbent, the method of its modification and the type of compound being removed. The achieved adsorption equilibrium is expressed by the relationship describing the change in the amount of adsorbed substance over time.

The dependence of the final adsorbate concentration for WDA and WD (HCl) on adsorption is described in Figure 3.

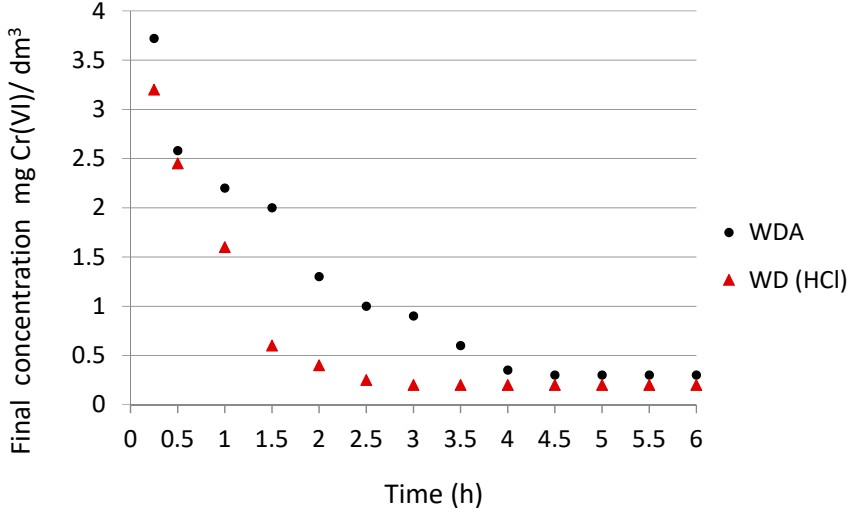

**Figure 3.** Chromium (VI) final concentration in solution at pH 2.

From the obtained test results, it can be concluded that the adsorption time had a significant effect on the chromium (VI) removal efficiency of a particular carbon. The maximum reduction of chromate ions with respect to WDA occurred after 4.5 h. The rate of chromium removal on WD (HCl) carbon was higher and the adsorption equilibrium time occurred after 2.5 h. Then stabilization occurred and for further tests (determination of isotherms), retention times of 2.5 (WD (HCl)) and 4.5 (WDA) hours were assumed. The results of many authors indicate that the contact time for different carbons usually does not exceed 6 h [22,24,29].

### 3.4. Adsorption Isotherms

The adsorption process was characterized on the basis of Freundlich adsorption isotherms determined at 288.15, 313.15 and 338.15 [K], at pH 2, for both adsorbents.

Freundlich isotherms were presented in a linear form and on their basis constants *K* and *n* were determined.

The Freundlich equation in the linear form has the following form:

$$log\frac{X}{m} = logK + \frac{1}{n}logC \tag{2}$$

where:

→　*X*—amount of adsorbed substance [mg]
→　*M*—adsorbent mass [g]
→　*C*—equilibrium concentration [mg/dm$^3$]
→　*K*, *n*—isotherm constants

The chromium(VI) adsorption isotherms for tested carbons are shown in Figures 4 and 5.

Tables 2 and 3 present isotherms constants *K*, *n* and the degree of matching (coefficient of determination R$^2$) to experimental conditions.

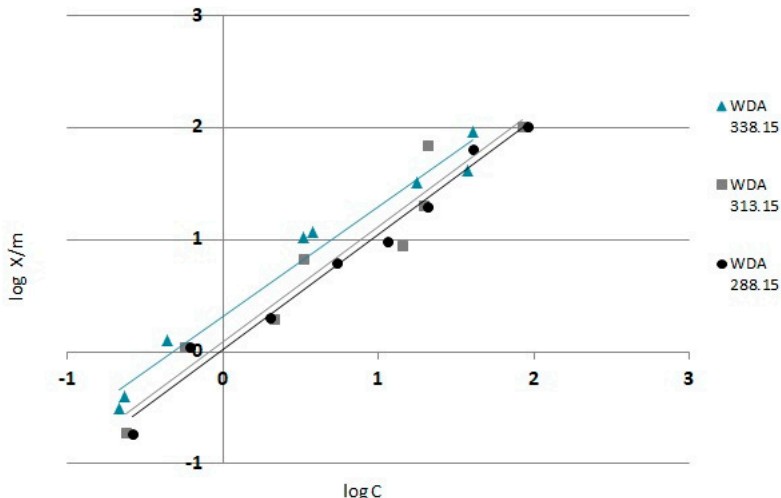

**Figure 4.** Freudlich adsorption isotherms for WDA, at temperatures: 288.15; 313.15; 338.15 [K].

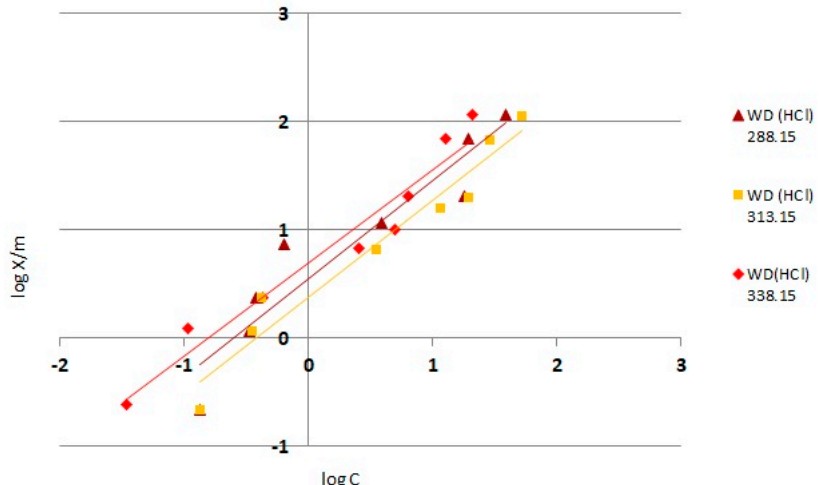

**Figure 5.** Freudlich adsorption isotherms for WD(HCl) at temperatures: 288.15; 313.15; 338.15 [K].

The analysis of the isotherms allows to conclude that matching the Freundlich mathematical model to experimental values was very good for both carbons, regardless of the solution temperature, as evidenced by high determination coefficients $R^2$ (above 0.9).

Comparing the values of coefficients *K*, it can be concluded that they were much higher for WD (HCl) modified carbon, which indicates the possibility of obtaining larger adsorption capacities for the same equilibrium concentrations. According to literature data, these values are comparable with other activated carbons adsorbing chromium Cr (VI) compounds [11].

**Table 2.** Values of constants *n* and *K* of Freudlich isotherms for WDA.

| Temperature | N | K | Coefficients $R^2$ |
|---|---|---|---|
| 288.15 K | 0.972 | 1.035 | 0.97 |
| 313.15 K | 0.973 | 1.238 | 0.93 |
| 338.15 K | 1.028 | 2.032 | 0.97 |

**Table 3.** Values of constants *n* and *K* of Freudlich isotherms for WD (HCl).

| Temperature | N | K | Coefficients $R^2$ |
|---|---|---|---|
| 288.15 K | 1.179 | 2.454 | 0.94 |
| 313.15 K | 1.106 | 3.589 | 0.90 |
| 338.15 K | 1.157 | 5.073 | 0.95 |

For WDA and WD (HCl) a slight temperature effect on the removal of chromium was noticeable, however, along with its growth, the adsorbents showed better adsorption properties. For both tested carbons, the largest adsorption capacities were obtained for the highest temperature of 338.15 K. In turn Rai et al. [29] found that for activated mango kernel the sorption capacity slightly decreased with the increasing temperature. Analyzing the obtained isotherms, it can be concluded that the chromium (VI) adsorption on WD (HCl) was much better. This is clearly confirmed by constant values *n* and *K*. The maximum capacity was $P_{max\ WD\ (HCl)}$ = 114 mg Cr (VI)/g.

Comparing WD (HCl) (under static conditions) with other carbons, it showed a good Cr(VI) adsorbent, sometimes achieving adsorption capacities several times higher [22–24,27,29].

### 3.5. Adsorption Dynamics

Adsorption under flow through conditions is carried out on a filter bed at specified parameters. Deterioration of the assumed adsorbent adsorption capacity is defined as the breakthrough of the bed. It ceases to absorb molecules, and the adsorbate concentration increases in the filtrate. The process continues until the bed is fully saturated and reaches the point of exhaustion. The adsorption capacity of the adsorbent layer to the breakthrough point and exhaustion is determined by such process factors as: Adsorbent type, initial concentration $C_0$ of the adsorbed component and contact (retention) time $T_R$. The breakthrough time for a given adsorbent decreases with decreasing $T_R$ and increase in $C_0$ while the adsorption capacity increases for higher initial concentrations of $C_0$ adsorbate, as illustrated by adsorption isotherms (Figures 4 and 5).

The filtration process to the point of bed depletion for WDA and WD (HCl) carbons, with contact time $T_R$ = 420 s, filtration rates v = 2 m/h and initial concentration $C_0$ = 1 mg/dm$^3$ is illustrated by Figure 6.

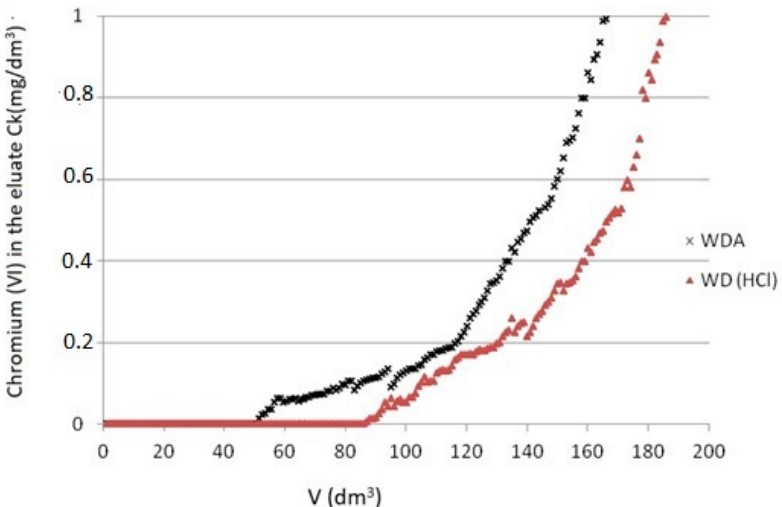

**Figure 6.** Adsorption isoplanes of Cr (VI) for WDA and WD (HCl), $C_0$ = 1 mg/dm$^3$, $T_R$ = 420 s.

Adsorption of chromium on the tested beds varied. The chromium in the filtration column with the WDA carbon was removed to a lesser degree. The filtration was shorter and only the initial removal of chromium (VI) was obtained in the initial phase. The concentration of chromium in the spill to the

bed breakthrough point fluctuated in range $C$ = 0.000–0.008 mg/dm$^3$. The adsorption capacity at the bed breakthrough point (adopted after filtering 50 L of solution) was $P_{bWDA}$ = 1.38 mg/g.

The WD (HCl) bed worked much more efficiently. The filtration cycle to the bed breakthrough point lasted twice as long, and the chromium compounds were removed almost completely ($C$ = 0.000–0.002 mg/dm$^3$). At the bed breakthrough point, adopted after filtration of 90 litres of adsorption, adsorption capacity $P_{b\ WD(HCl)}$ = 2.25 mg/g as obtained. For comparison, the adsorption capacity for another bed (mixed activated carbon) was set at 6.67 mg/g but with a much higher Cr(VI) concentration (40 mg/dm$^3$) [30].

The calculated adsorption capacities after exhaustion of the bed were, respectively: $P_{eWDA}$ = 4.02 mg/g and $P_{eWD\ (HCl)}$ = 4.35 mg/g. Referring these values to the results obtained under static conditions ($C_0$ = 10 mg/dm$^3$) (Figure 2), it was found that dynamic adsorption capacities are much higher. This relationship is also confirmed by other studies of the adsorption process [31]. To increase the efficiency of dynamic conditions, tests should be continued, optimizing the retention time for various initial concentrations, because on a technical scale granular activated carbon is used in the filtration process (as an adsorption bed).

## 4. Conclusions

- The solution pH significantly affected the adsorption capacity of the tested activated carbons. For the WD carbon (HNO$_3$) the adsorption decreased slightly with increasing the solution pH. For carbons WDA and WD (HCl), the effect of the reaction on the chromium adsorption capacity was greater and definitely more beneficial at a lower pH.
- The action of inorganic acids (chemical modification) on activated commercial carbon WD-extra caused a change in its sorption properties. The use of hydrochloric acid contributed to the increase in the adsorption capacity of chromium (VI), by increasing the specific surface area of carbon as a result of the mesoporous ash removal.
- Modification with nitric acid (oxidizing) caused a decrease in the adsorption capacity, most likely associated with a change in the chemical nature of the carbon surface and partial destruction of the pore structure due to carbon oxidation.
- The adsorption time had a significant effect on the efficiency of chromium (VI) removal by the specific coal. For carbon WD(HCl) the adsorption equilibrium occurred after 2.5 h. The maximum reduction of chromate ions with respect to WDA occurred after 4.5 h.
- Analyzing the determined isotherms, it can be concluded that as the temperature increased, the adsorbents showed better adsorption properties, whereas the highest adsorption capacity of chromium (VI) was demonstrated by WD(HCl) modified carbon.
- Adsorption under flow through conditions showed that a modified WD(HCl) carbon bed worked much more efficiently than WDA. The filtration cycle to the bed breakthrough point lasted twice as long, and the chromium compounds were removed almost entirely. Adsorption capacities were obtained $P_{bWD(HCl)}$ = 2.25 mg/g (breakthrough point) and $P_{eWD\ (HCl)}$ = 4.35 mg/g (after exhaustion).
- In the light of the carried-out research, the modified WD (HCl) carbon effectively removed chromium (VI) compounds from water, which makes it possible to use it in water treatment systems.

**Author Contributions:** A.P. conceptualization and methodology; A.P. and J.K. performed the adsorption experiments; A.P. formal analysis; A.P. and J.K. carried out the measurements and calculations; A.P. and J.K. writing—review and editing.

**Funding:** This scientific work was implemented within the frame of the project "Maintaining research potential of the Rzeszów University of Technology", number: DS.BO.19.001.

**Conflicts of Interest:** The authors declare no conflict of interest.

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
