# Peer review of "Chromium (VI) Adsorption on Modified Activated Carbons"

_applsci, doi:10.3390/app9173549_

Round 1

Reviewer 1 Report

In this work, the authors investigated Chromium adsorption over three different carbons. Batch and continuous experiment was performed, and the data were fitted by empirical models. This work could provide valuable info in the field of water treatment, however some issues need to be addressed before acceptance for publication.

There are already multiple publications which investigated Chromium adsorption over carbon. The authors should include those works in the Introduction, compare the difference, and address the novelty of the current work.

In Figure 2, what is the experimental condition (i.e. pH)? Also, did the authors do any duplicated runs? In order to make a strong conclusion on the comparison of the performance of the three adsorbents, experimental error bars are necessary.

I suggest the authors characterize the physical properties of the modified carbons, compare the structure with the unmodified carbon, and use the result to support their hypothesis regarding the different adsorption capacity of the three carbons.

From Figure 4 to 6, please colore the trend lines, so we can know which trend line is for which adsorbent.

Author Response

Please see the attachmant

Reviewer 2 Report

The subject of this paper is interesting, but it lacks a lot of analysis, and the following must be revised before reconsideration.

Authors should unify the terms for chromium, such as chromium (VI), Cr(VI), Cr6+, and chromate. Keywords are not enough to cover the whole contents. Literature review of relevant studies and description of originality of this study must be added to the introduction. The figures must be redrawn in the same form. Unify the units into SI units. State clearly the source of WDA, hydrochloric acid, and nitric acid used in the experiment. The changes in activated carbon after treatment must be characterized by BET and FTIR analysis as a minimum. (Line 136) What is the meaning of “so significant”? (Line 133) “adsorption decreased slightly with increasing the solution pH”

(Line 153) "increased the specific surface area of mesopores in the carbon."

(Line 160) “increasing the polarity and hydrophilicity of the surface”

Please provide data to support the above arguments.

What is the reaction time for pH test? Figure 6 should be deleted as duplicates. (Line 257) “190 litres of adsorption” should be changed to 90 liters.

Round 2

Reviewer 2 Report

Although the revised paper has improved a lot, it is still not well organized for the reader.

1) The revised parts in introduction section must be rewritten. Authors need to be careful of typos. These things reduce the quality of the entire paper.

2) What does "WD" mean?

3) The novelty statement of this paper is still unclear.

4) All figures must be redrawn to make it easy understanding with unified forms (especially Figure 2).

5) It is unbelievable that nitrogen adsorption for BET analysis was performed at 200 degrees. Authors should add a detailed description of the analytical method to the materials and method session.

6) Overall, there is a lack of discussion. Authors should additionally compare the results with the relevant literature.

Round 3

Reviewer 2 Report

It can be published in present form.